# Horizontal Tensile Machine for Mechanical Tests Applicable to Suspension Clamps, Transmission Line Accessories, and Overhead Conductors

Antonio Ramirez-Martinez *, Leonardo Barriga-Rodriguez, Noe Amir Rodríguez-Olivares and Jorge Alberto Soto-Cajiga

CIDESI, Centro de Ingeniería y Desarrollo Industrial, Dirección de Ingeniería Eléctrica y Electrónica, Av. Playa Pie de la Cuesta 702, col. Desarrollo San Pablo, Querétaro 76125, CP, Mexico; lbarriga@cidesi.edu.mx (L.B.-R.); noe.rodriguez@cidesi.edu.mx (N.A.R.-O.); jsoto@cidesi.edu.mx (J.A.S.-C.)
* Correspondence: armartinez@cidesi.edu.mx; Tel.: +52-442-2119800

**Abstract:** This work aimed to design a tensile horizontal machine that performs mechanical testing for suspension clamps, transmission line accessories, and overhead conductors with the following features: the suspension clamps device will be a permanent part of the structure, requiring a minimal setup; and it will accept overhead conductor specimens with lengths of up to 12 m and also be able to perform testing for pieces that require the plate-fork fastening option. Analytical and numerical calculations are performed according to the AISC-360-16 standard and static structural module of ANSYS software, respectively, to compare results.

**Keywords:** suspension clamps; IEC 61395 standard; fitting; overhead conductor; IEC 61089 standard

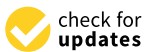



## 1. Introduction

The universal vertical tension-compression machine (VTM) of the mechanical testing laboratory plays an essential role in the stress-strain characterization tests of materials; the average stroke length of commercial machines ranges from 1200 mm to 1500 mm; this range is insufficient to perform this type of test for some accessories used in the construction of power transmission lines. Although a vertically arranged machine can perform the test for most of the elements of this group, two elements are an essential part, whose requirements of the standards that regulate them require horizontal tensile machines (HTMs).

These elements are the overhead transmission conductor and the suspension clamps, for the tests of the first one; the use of an HTM is indispensable according to the requirements of the standard for Aluminum Conductors in Overhead Lines ACSR and IEC norm because the length of the specimens reach up to 12 m. In the case of the second element, the actual test uses a VTM, which presents some inconveniences that can eliminate them by using an HTM.

To complete the versatility of the proposed machine, elements must be included that use the traditional fork-plate gripping method, avoiding major adjustments or preparations.

For the case of mechanical testing of suspension clamps, Figure 1a,b shows a couple of arrangements proposed by the international standard [1] (p. 59), in which arrangement 1(b) is that which has the most significant similarity with the patents found in the state of the art [2–4]. These devices are not an integral part of this type of machine, so it is mandatory to assemble-disassemble the device to perform tests of specimens different from the suspension clamps. Among the disadvantages is that, due to the tensile test loads of the specimens, the structure of the device is heavy, making it difficult to be handled directly by human personnel; this feature makes necessary the use of machinery, such as forklifts, to place it between the supports of the machine to position and hold it, with the potential risk of collision with the structure of the device.

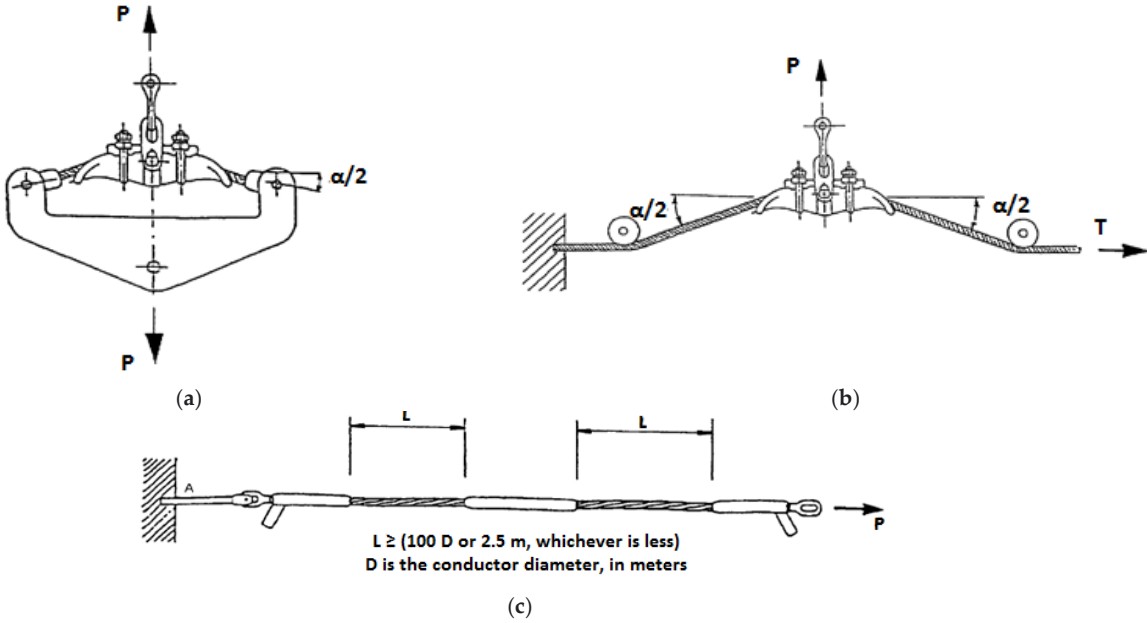

**Figure 1.** (**a**) Arrangements for suspension clamp mechanical testing; (**b**) second option; and (**c**) sliding test arrangement layout.

This standard also regulates the slip test; Figure 1c shows a combination of various types of fixtures, where arrangements such as this result in considerable specimen lengths, which, as mentioned, for a vertically arranged machine, the test stroke does not allow performing the test.

For the case of overhead conductor testing, referring to the topic of HTMs, in state of the art, a few works can be found, and the topics found refer to the testing of overhead conductors in the transient creep test [5]. The scope refers to a tensile test range from 15% to 35% of the cable's RTS (rate tensile strength) value under testing [6] for a specimen length of up to 12 m. After completion of this test, the stress-strain test is necessary for complete mechanical characterization, in which loads can reach up to 200 kN, so the design of an HTM with a capacity of up to 300 kN is analyzed.

Some standards applicable to overhead conductors are the Aluminum Association [7] and IEC 61089 [8], and one of their requirements indicates that the test specimen should reach up to 12 m.

The use of long-length specimens is not only required for overhead conductors; it is also applicable for conductors, such as OPGW (optical ground wire), that require specimens of 25 m in minimum length for tensile performance, according to IEC 60794 [9] (p. 12). However, there are standards such as ASTM D 3039 [10] (p. 6) that specify 250-mm length specimens for the elongation-failure test for the same OPGW cable; due to this dimension, running it in a VTM does not represent any problem, as in the research performed in [11].

It is important to note that using a horizontal machine allows for the use of specimens of long lengths, translating into a more representative sample of the quality of the product.

By the requirements of the referenced standards regarding the length of the overhead conductor specimens and the particular application of the suspension clamp, this work focuses on the development of a horizontal tensile machine with a load capacity of up to 300 kN, which facilitates the performance of mechanical tests for the whole group of fittings.

## 2. Materials and Methods

### 2.1. Fundamentals

A structural member has been subjected to compression if its design primarily resists an axial compression load, although some bending may also be present and accounted for

in the design. If the bending action is significant, the member is called a beam-column and is designed differently.

Compression members can fail by yielding or elastic buckling, depending on the slenderness ratio of the members, as well as by local buckling. Members with low slenderness ratios (short columns) generally fail by yielding (crushing or squashing). In contrast, the failure tends to occur by elastic buckling for members with high slenderness ratios (slender columns).

Slenderness Limits

The ratio of the effective length of a typically compressed member to the radius of gyration, both referred to the same bending axis, is called the slenderness ratio.

With the slenderness ratio of a typically compressed member, the length will be its effective length $kL$, and $r$ will be the corresponding radius of gyration. The slenderness ratios $kL/r$ of compressed members shall preferably not exceed 200.

Table 1 is an excerpt from [12] (pp. 17–19), Table B4, specific to HSS (hollow structural sections) discussed in this document.

**Table 1.** Excerpt from table B4.1b of AISC-360-16 standard.

| Description of Element | Width-to-Thickness Ratio ($\lambda$) | $\lambda_p$ (Compact/Non-Compact) | $\lambda_r$ (Slender/Non-Slender) | Examples |
|---|---|---|---|---|
| Flanges of rectangular HSS | b/t | $1.12\sqrt{\frac{E}{F_y}}$ | $1.40\sqrt{\frac{E}{F_y}}$ | |
| Webs or rectangular HSS and box sections | h/t | $2.42\sqrt{\frac{E}{F_y}}$ | $5.70\sqrt{\frac{E}{F_y}}$ | |

Table 2 shows the relationships for category classifications of sections and structural elements regarding slenderness.

**Table 2.** Classification of elements and cross-sections in profiles.

| Classification of Profiles by Compression | | | Classification of Profiles by Buckling | | |
|---|---|---|---|---|---|
| If | $\lambda \leq \lambda_r$ | non slender element | If | $\lambda < \lambda_p$ | compact section |
| If | $\lambda > \lambda_r$ | slender element | If | $\lambda_p \leq \lambda \leq \lambda_r$ | non compact section |
| | | | If | $\lambda > \lambda_r$ | slender section |

Since the tests are based only on tensile load tests, meaning that the structure will support compression loads, and adding the dimensions of the specimens, it is necessary to focus the analysis on compression and buckling phenomena in structures, calculating the nominal compressive strength, which is based on the limit state of flexural buckling:

$$P_n = \varnothing_c \, F_{cr} A_g \tag{1}$$

In addition to calculating the gross area $A_g$ of any element, another parameter to consider for calculating the maximum force to be applied is the net area $A_n$, mainly when there are holes, which reduce the pressure to be applied. According to [12] (p. 16.1–20), for elements containing a chain of holes extending across a face in any diagonal or zigzag line, the net width of the part is obtained by subtracting from the gross width the sum of the diameters or slot dimensions, as given in this section, of all holes in the chain and adding, for each gauge spacing of the chain, the quantity $\frac{s^2}{4g}$.

Table 3 contains the equations to calculate the critical stress for elements subject to compression, considering the width-to-thickness ratio according to [12] (pp. 35–36), and Table 4 defines the limits for common $F_e$ [12] (p. 311).

**Table 3.** Critical stress compression formulas.

| The Critical Stress, $F_{cr}$, Is Determined as Follows | |
| --- | --- |
| (a) if $\frac{L_c}{r} \leq 4.71\sqrt{\frac{E}{F_y}}$ | (b) if $\frac{L_c}{r} > 4.71\sqrt{\frac{E}{F_y}}$ |
| $\left(\text{or } \frac{F_y}{F_e} \leq 2.25\right)$ | $\left(\text{or } \frac{F_y}{F_e} > 2.25\right)$ |
| then $F_{cr} = \left(0.658^{\frac{F_y}{F_e}}\right)F_y$ | then $F_{cr} = 0.877F_e$ |
| Where: $F_e = \dfrac{\pi^2 E}{\left(\frac{L_c}{r}\right)^2}$ | |

**Table 4.** Limits defining $F_e$.

| TABLE C-E3.1 Limiting Values of $\frac{L_c}{r}$ and $F_e$ | | |
| --- | --- | --- |
| $F_y$ Ksi (MPa) | Limiting $\frac{L_c}{r}$ | $F_e$ Ksi (MPa) |
| 36 (250) | 134 | 16.0 (110) |
| 50 (345) | 113 | 22.2 (150) |
| 65 (450) | 99.5 | 28.9 (200) |
| 70 (485) | 95.9 | 31.1 (210) |

Figure 2 shows the comparison plots of the AISC-360 standard and Euler's buckling of elements subjected to compression loads, the former of which is more conservative.

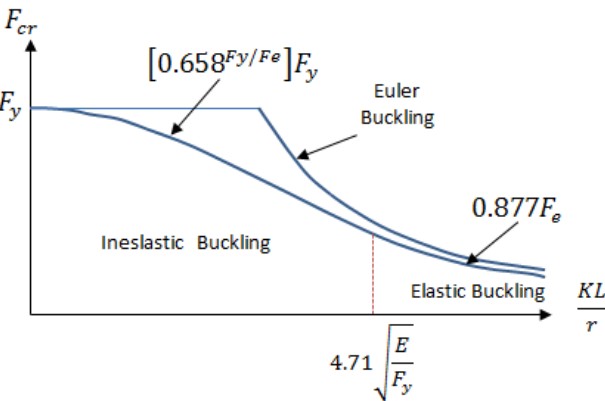

**Figure 2.** Column buckling curves.

Table 5 shows the calculations based on the limit states and the type of element cross-sections subjected to compressive loading [12] (p. 34). According to the referenced standard for HSS, the section "E3. Flexural Buckling of Members without Slender Elements" or "E7. Members with Slender Elements" is applicable, depending on the type of slenderness.

**Table 5.** Limit states for cross-section profiles.

| TABLE USER NOTE E1.1 Selection Table for the Application of Chapter E Sections | | | | |
|---|---|---|---|---|
| | **Without Slender Elements** | | **With Slender Elements** | |
| **Cross Section** | **Sections in Chapter E** | **Limit Sates** | **Sections in Chapter E** | **Limit Sates** |
| ⬒ | E3 E4 | FB TB | E7 | LB FB TB |
| ⬐ ⬒ ⬓ | E3 E4 | FB FTB | E7 | LB FB FTB |
| ▯ | E3 | FB | E7 | LB FB |

### 2.2. Machine analysis

For the proposed servo-driven horizontal tensioning machine, there are two main configurations in Figure 3, in which the difference is the number of ball screws, which influence the tensile load capacity. The load capacity and the length of the specimen determine the required robustness of the structure, which in turn determines the selection of the rail size.

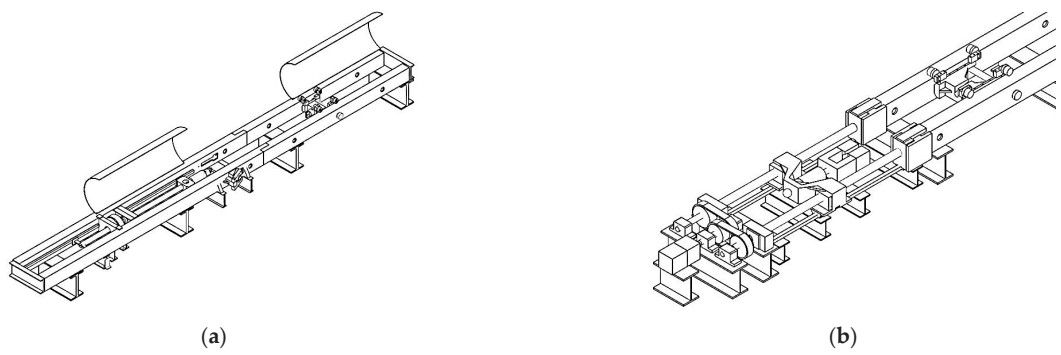

(**a**)            (**b**)

**Figure 3.** Horizontal tensile machines: (**a**) one ball screw; (**b**) two ball screws.

As mentioned above, buckling is a phenomenon that requires special attention in the design of long and slender structures. For column construction, built-up elements from single rolled sections usually improve the element's strength along its axes, and this technique applies to this work. The research focused on the study of simple [13] and built-up [14] profiles subjected to compression and combined forces based on Japanese standards agrees that the slenderness ratio is reliable for the beginning of stress design analysis, in agreement with the AIS 360-16 standard, which is the basis for the analytical study of this work.

Regarding the set of fittings offered by the manufacturers, according to their catalogs [15,16], it is observed that 95% of the parts have a load capacity of less than 200 kN, and almost the same percentage use the fork-plate system for specimen clamping. Therefore, the clamping system is mandatory in the HTM to give it greater versatility.

Table 6 shows the properties and dimensions of profile C10 × 30, which is the material used in this research.

**Table 6.** Properties and dimensions of C profile C10 × 30.

| Height ×<br>Weight | Properties | | | | | | Dimensions | | | | | |
|---|---|---|---|---|---|---|---|---|---|---|---|---|
| | Dimensions | | | | | | Properties | | | | | |
| | d<br>(mm) | $t_w$<br>(mm) | Flange | | m<br>(mm) | T<br>(mm) | Axis X-X | | | Axis Y-Y | | |
| | | | Width $b_f$<br>(mm) | Thickness $t_f$<br>(mm) | | | A<br>(cm²) | I<br>(cm⁴) | r<br>(cm) | I<br>(cm⁴) | r<br>(cm) | x<br>(cm) |
| C10×30 | 254 | 17.1 | 77.0 | 11.1 | 25.4 | 203 | 56.9 | 4287 | 8.69 | 164 | 1.70 | 1.65 |

The set of HTM systems is similar to VTM, and they are classified as follows:

(a) Traction system: spindle, packed guides, pulleys, and servo-reducer;
(b) Clamping system: forks and plates;
(c) Test stroke length;
(d) Tailstock or movable carriage;
(e) Welded mechanical structure;
(f) Suspension clamp device.

As stated, the analysis focuses on three elements that constitute the main difference from the VTM; the first is the integrated device for suspension clamps, the second is the traveling carriage, and the third one is the structure.

2.2.1. Suspension Clamp Device

The suspension clamp testing device comprises a suspension clamp holder tool and a support block assembly. The clamp holder includes a steel cable and two steel plates at one end, joined together by four bolts, the objective of which is to assist in fine adjustments to precisely attain the value of the clamp dangle angle. At the steel cable's ends, cylindrical tin-cast blocks anchor onto the support blocks placed on the machine's exterior web profile during the test performance. It is necessary to use steel cable for the test to emulate the force-stress of the line and the suspension clamps during their operation, according to the study performed in the investigations [17,18].

The support blocks comprise an H-shaped plate containing a slot that allows for passing through the steel cable to position it on the machine structure. The other blocks are those that provide the dangle angle.

Figure 4a shows the suspension clamp device holder, constructed of a flexible steel cable and rectangular plates joined together by four bolts. Figure 4b shows the layout for the mechanical test suspension clamps.

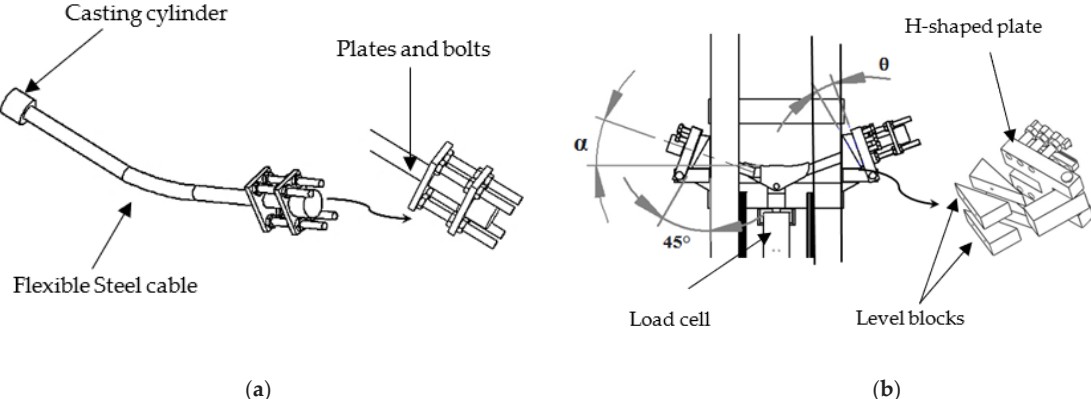

(**a**)
(**b**)

**Figure 4.** (**a**) Suspension clamps holder; (**b**) arrangement for mechanical suspension clamp test; dangle angle α and adjustment angle θ.

### 2.2.2. Movable Carriage

The proposed tensile machine has a movable carriage comprised of one circular bar, two bushings, and one nut, as shown in Figure 5a, and it is moved manually along the length of the rail. The traveling carriage's position depends on the length of the specimen tested. The round bar passes through the web rail to create the anchorage. It is necessary to use a pair of bushings, length of which is the rail width, and the purpose of which is to distribute the stresses on the profile's web, with the nut ensuring the position.

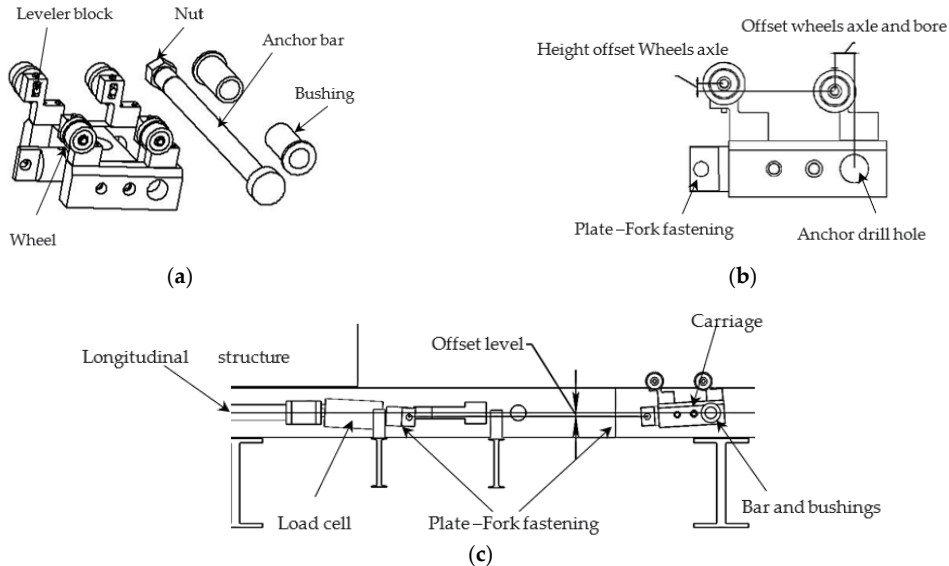

(**a**)
(**b**)

(**c**)

**Figure 5.** (**a**) Elements and parts of the carriage; (**b**) side view of carriage; and (**c**) view of the load cell and carriage initial arrangement.

The traveling carriage has four leveler blocks, the structure of which has a groove to accommodate the wheel axle and three bores distributed along its thickness. The wheel axle, at one end, has three holes that match the holes in the block—one threaded hole located in the middle of the two through holes. The threaded hole allows for moving the axle along the groove using a screw placed at the central bore of the wheel block. Regarding wheel stability, placing a couple of bars into the two bores' extreme axle prevents gyration. It allows the axle to be positioned at any height, limited by the groove length in the block.

Regarding the level of the carriage wheels, it is essential to align the longitudinal axis of the carriage and the load cell, preventing the rear wheels from colliding with the wall of the rails during the tensile load test and interfering with the measurement of the load

magnitude. To achieve the alignment of the elements, the axle of the rear wheels should be offset toward the front wheels so that the diameter of the wheel is tangential to the diameter of the anchor bolt, as shown in Figure 5b.

Also, placing the front and rear axles of the mobile carriage wheels at different heights—the front ones lower relative to the floor—allows for maintaining a position below their longitudinal axis. The tilting position is achieved for the load cell by placing a support to hold it in that position, as shown in Figure 5c. Both rise when the tensile load is applied, obtaining collinear alignment. This position has two advantages: the first is that it allows the load cell to record a correct measurement; and the second is that the wheels of the cart do not touch the wall of the rails, leaving the carriage suspended only from the anchor bar, ensuring that the wheels will not be damaged and will have long use.

To ensure that the load cell will have prolonged use, it must have a ball joint that provides three degrees of rotational freedom to absorb misalignment.

### 2.2.3. Structure

The particular requirements to be met by the structure:

- Tensile load capacity up to 300 kN;
- Testing of specimens up to 12 m in length according to the standards [7,8];
- Having suspension clamps integrated into the structure, which requires minimal set-up;
- Ability to include tests in which the clamping for the specimens can use a fork-plate.

The structure consists of rails comprising two C-profile types (CE) welded on the flange side, with a total length of 17.5 m for a 1.5-m stroke length. The rails rest on 18" × 7 1/2" IPR profiles, with a group brace length of 3000 mm. The rails have 80 mm-diameter through holes drilled 1 m apart and placed on the web profile side, aiming to anchor the carriage using the bushings and the bar.

Also, the structure has two manufactured T-shaped slots on the web side rails, allowing one end of the flexible device to pass through both rails for the suspension clamp test. To prevent the structure from being weakened by the fabricated slots and to resist the lateral forces generated mainly by the test, a pair of IPR sections are added and welded to the ends of the slot length parallel to the bases. Additionally, a couple of 1"-diameter steel bars are added inside profiles—one placed at the bottom and another at the top half the length of the slot—ensuring robustness, as shown in Figure 6.

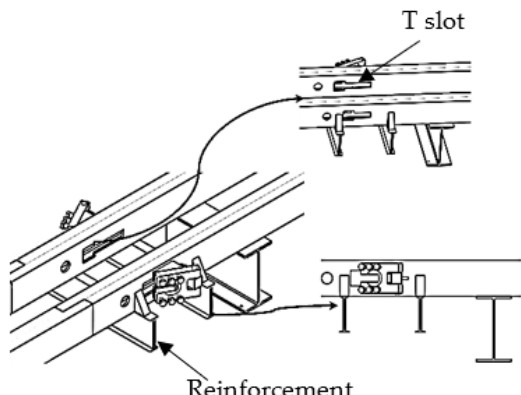

**Figure 6.** Views of grooves and reinforcement of suspension clamp device.

The analytical calculation using the AISC 360-16 standard only applies to the test for the overhead conductor. For the case of the device for suspension clamps, this standard is not applicable since it does not apply Euler's law. In the case of the numerical method, it applies to both tests.

## 3. Results

### 3.1. Analytical Method

Figure 7 shows the cross-section profile built up (a) and a table (b) containing the inertia and gyration radius using the formulas of Steiner's theorem; $I = I' + dA^2$ and gyration radius; $r = \sqrt{\frac{I}{A}}$.

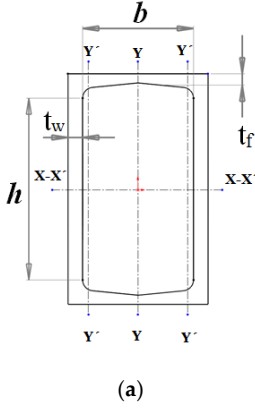

| Properties | | | | |
|---|---|---|---|---|
| | Axis X-X | | Axis Y-Y | |
| A | I | r | I | r |
| (cm²) | (cm⁴) | (cm) | (cm⁴) | (cm) |
| 154 | 8574 | 8.69 | 4493 | 6.28 |

(**a**)        (**b**)

**Figure 7.** Built-up profile member: (**a**) cross-sectional; (**b**) inertia moment and radius gyration values.

The relationships in Table 2 show that the structure is compact and does not contain slender elements; only the flexural buckling calculation applies. Additionally, the buckling mode for this structure does not involve relative deformations that produce shear forces in the connectors between individual shapes; therefore, it is calculated as stated in section E3 [12] (p. 16.1–35).

Table 7 shows the values resulting from the data in Table 6.

**Table 7.** Values by applying the dimensions of the C profile.

| Description Element | $(\lambda)$ | $\lambda_p$ | $\lambda_r$ | Description Element | $(\lambda)$ | $\lambda_p$ | $\lambda_r$ |
|---|---|---|---|---|---|---|---|
| Flanges of rectangular HSS | $b/t_f = 10.87$ | 32.26 | 40.33 | Webs of rectangular HSS | $h/t_w = 11.94$ | 69.72 | 164.21 |

The relationships in Table 2 show that the structure is compact and does not contain slender elements; only the flexural buckling calculation applies for analysis. Additionally, the buckling mode for this structure does not involve relative deformations that produce shear forces in the connectors between individual shapes; therefore, it is calculated as stated in section E3 [12] (p. 16.1–35).

Table 8 shows the slenderness values from applying the data in the profile, as shown in Figure 7.

**Table 8.** Values of slenderness of the built-up profile.

| Slenderness Calculation with Respect to X-X Axis | | | | | Slenderness Calculation with Respect to Y-Y Axis | | | | |
|---|---|---|---|---|---|---|---|---|---|
| K | *L* (cm) | $L_{cx}$ (cm) | $r_x$ (cm) | $\lambda$ | K | *L* (cm) | $L_{cy}$ (cm) | $r_y$ (cm) | $\lambda$ |
| 1 | 1200 | 1200 | 15.38 | 78 | 0.7 | 300 | 210 | 6.8 | 33.43 |

According to Table 4 [12] (p. 311), the value of $F_e$ is limited to 110 MPa = 1121 kg/cm². This value governs the calculation of $F_{cr}$ to elaborate Table 9.

**Table 9.** $F_{cr}$ value according to Tables 3 and 4.

| The Critical Stress, $F_{cr}$, Is Determined as Follows | $F_e$ Adjusted | $F_{cr}$ Adjusted |
|---|---|---|
| (a) if $\frac{L_c}{r} \leq 4.71\sqrt{\frac{E}{F_y}}$ | $78 \leq 135.69$ | |
| then $F_{cr} = \left(0.658^{\frac{F_y}{F_e}}\right)F_y$ | 182.28 MPa | 96.56 MPa |
| where: $F_e = \frac{\pi^2 E}{\left(\frac{L_c}{r}\right)^2}$ | 334 MPa | 110 MPa |

Calculating the net area:

$$A_n = A_g - A_b \tag{2}$$

where $A_b$ corresponds to the bore area, which is taken from SolidWorks software 3D model.

$\varnothing_c = 0.90$; [12] (p.16.1–33).

From (2): $A_n = A_g - A_b = 113.8–85.451 = 28.348$ cm$^2$

Using (1): $P_n = \varnothing \, F_{cr} \, A_n = 273.821$ KN, net load that supports each rail.

According to the result, the machine can support a load of 547.643 KN, compared to the design load of 300 kN; a superior margin of 1.82 assures the machine's good performance and corroborates the proper selection of the built-up profile. The values obtained are valid only for overhead conductors and fitting tests and do not include the device for suspension clamps, which only applies the numerical method.

### 3.2. Numerical Method

The stresses and strain calculations of the structure for overhead conductors up to 12 m in length and the suspension clamps device tests apply the static structural module of ANSYS Workbench software for the analysis, applying a 300-kN load for both.

#### 3.2.1. Suspension Clamps Device Test

As mentioned above, the structural analysis for the clamp test device is validated only by the static structural modulus. Figure 8a shows the stress values, in which the range oscillates between 45 and 148 MPa. The value of 533 MPa is exactly at the bolt head area, which can be reduced significantly using high-quality steel (AISI 4340). Figure 8b shows the total deformation suffered by the structure, which is 1.1499 mm.

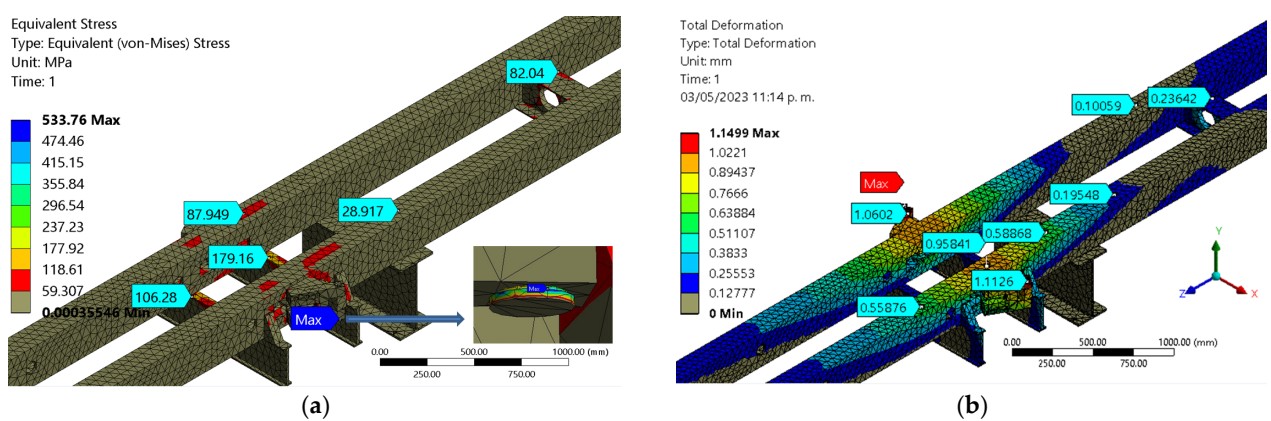

**(a)**　　　　　　　　　　　　　　　　　　　　　**(b)**

**Figure 8.** (**a**) Test stress values for suspension clamp device; (**b**) total deformation for suspension clamp device.

Figure 9a,b shows the deformations on the Z = 0.347 mm and X = 0.9907 axes. As expected, the X-axis value is the largest and is very close to the total deformation. According to these values, integrating the device into the HTM to test the suspension clamps does not affect the structural integrity.

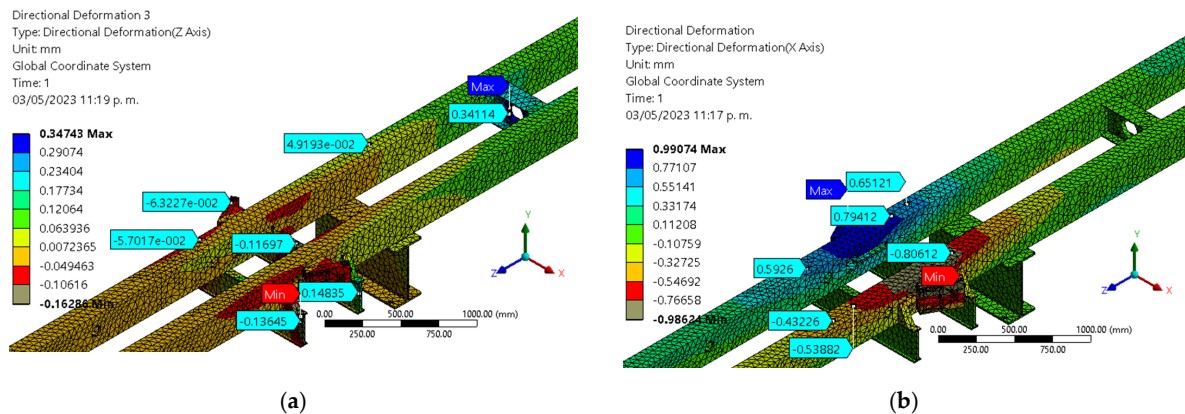

**Figure 9.** Directional deformation values: (**a**) Z axis; (**b**) X axis, for suspension clamp device.

### 3.2.2. Overhead Conductors

The calculation with the analytical method using the requirements of the AISC 360-16 standard indicates that the robustness of the machine structure for the tests of the overhead conductors is correct. The ANSYS software's static structural module is used to confirm that the results are acceptable. For this test, a critical element is the anchor bar, so special attention is paid to the results of this element. The material used in its manufacture is AISI 4340 standardized steel. The diameter of the bar is 60 mm.

Figure 10 shows the structural bar and beam stress values for a 12 m-long specimen with a tensile load of 300 kN. The beam system is built using the same C-profile.

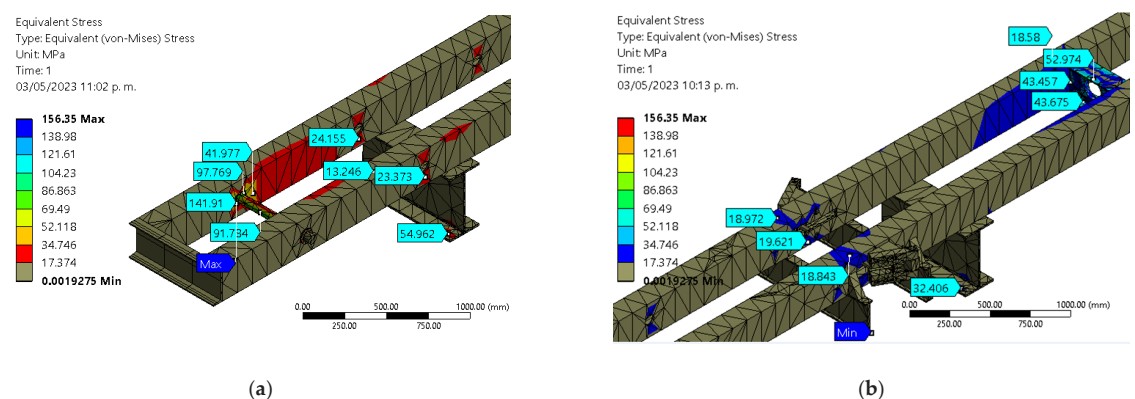

**Figure 10.** Stress values: (**a**) bar; (**b**) beam.

Figure 11 shows the values of: (a) the bar's total deformation = 0.95753; and (b) the bar´s safety factor = 1.599.

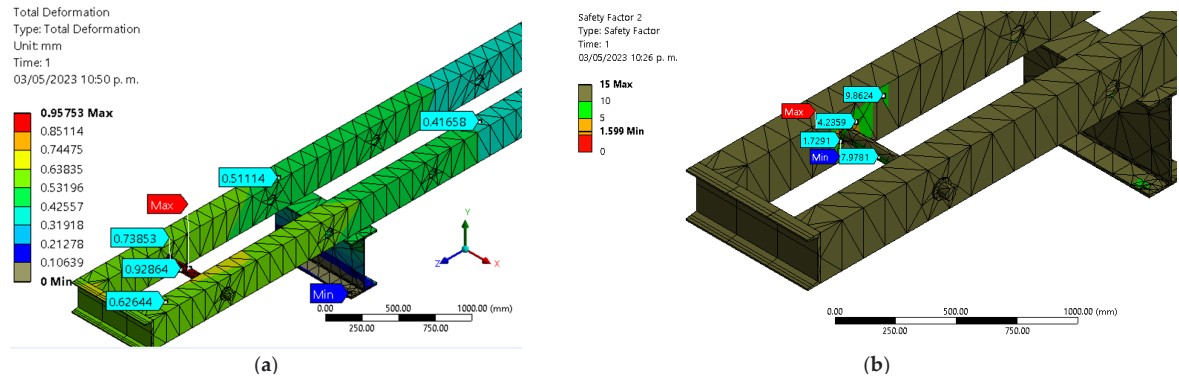

**Figure 11.** Bar´s values: (**a**) total deformation; (**b**) safety factor.

### 3.3. Manufacture

Figures 12–14 show the constructed horizontal tensioning machine, detailing the main contributions: the structure in Figure 13, the integration of the clamp testing device, in which the only element removed for a different test is the flexible steel cable device in Figure 13. Figure 14 shows the traveling carriage and load cell.

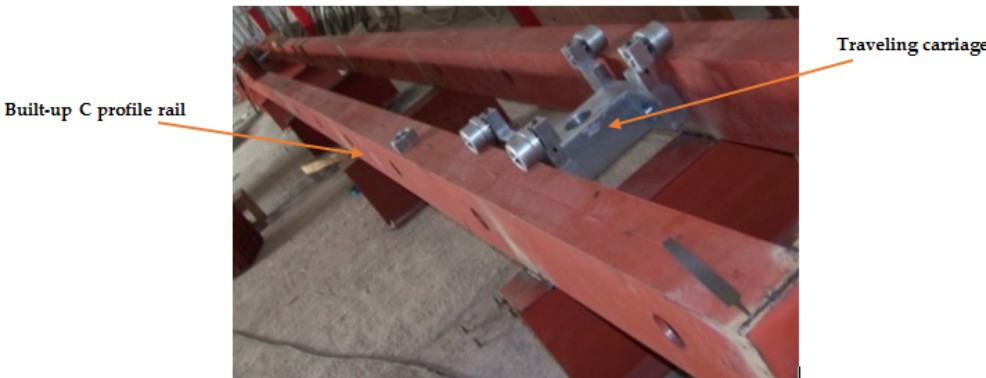

**Figure 12.** View of the construction of the HTM structure.

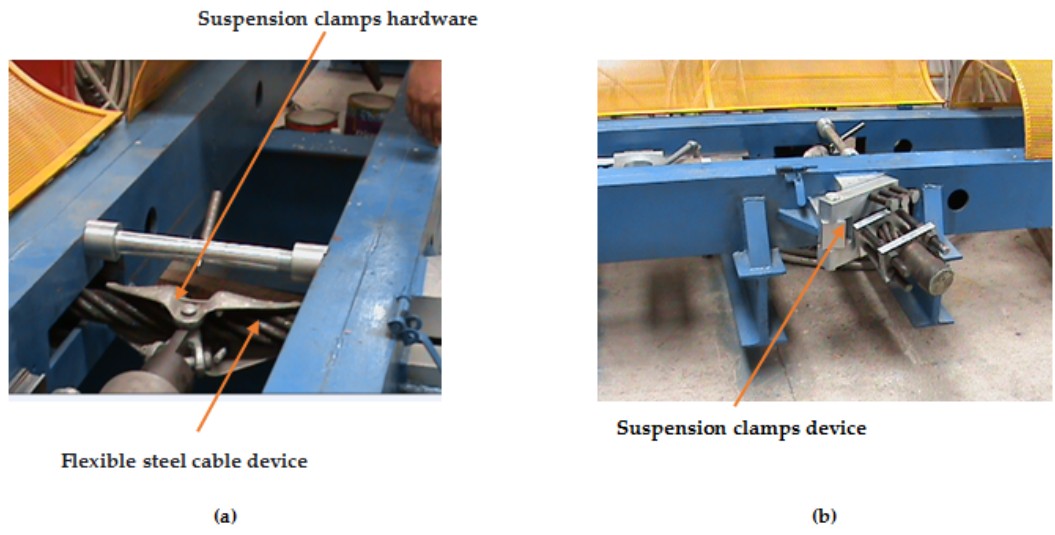

**Figure 13.** View of HTM: (**a**) suspension clamp hardware; (**b**) flexible steel cable device.

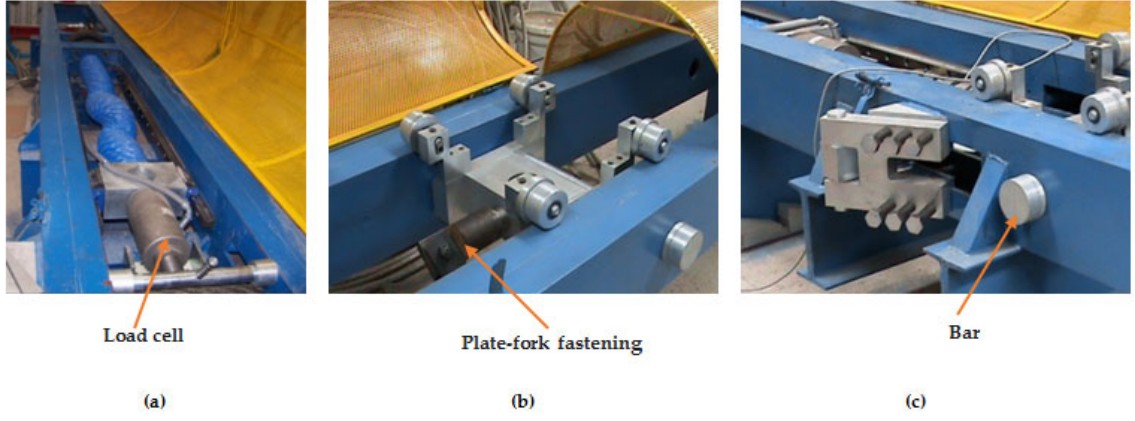

**Figure 14.** (**a**) Load cell view; (**b**) traveling carriage with plate-fork fastening element; and (**c**) traveling carriage at test position for short specimens.

### 3.4. Laboratory Test

The primary variable to control is the tension load, whose feed rate is 2500 N/min to 40,000 N/min. Monitoring and recording the load cell required a data acquisition system with analog inputs of +/−10 V with an 18-bit resolution.

To apply and initialize the control effectively, one can use a PID or PI, taking care that the tensile load should be at least 2% of the RTS, which ensures that the increments load does not have abrupt changes.

Figures 15 and 16 show the test load-failure curves of the suspension clamp and the overhead conductor, which compare the design against the working load. The loads are below 200 kN, 50% below the design capacity of the device.

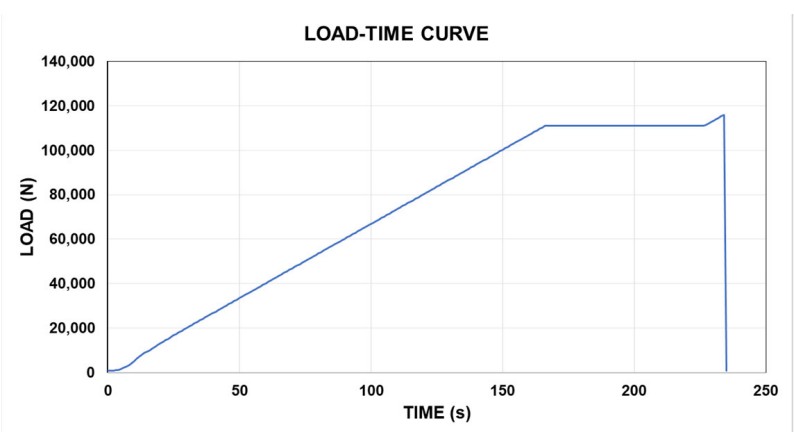

**Figure 15.** Failure load test curve for suspension clamps.

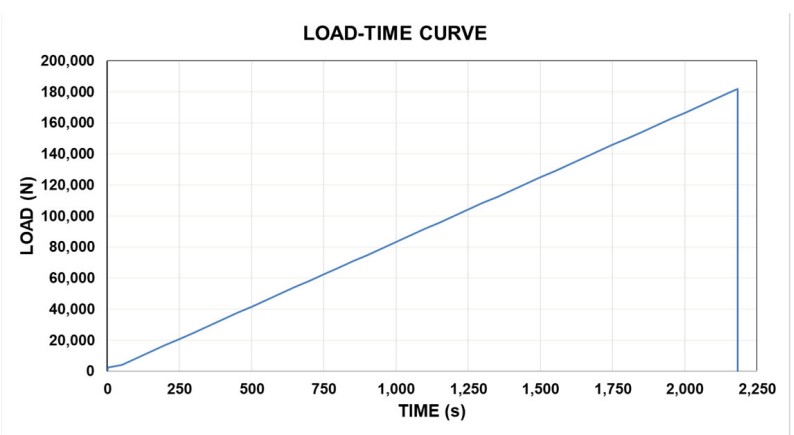

**Figure 16.** Failure load test curve for overhead.

The deformations experienced by the machine structure, longitudinal and lateral, concerning the laboratory tests, have an average variation of 43%, less than the numerical analysis of the design load (300 kN), measured with an analog indicator.

Figure 17 shows another stress-strain test performed on the overhead conductor according to CFE E1000-18 (page 17); this is done in several steps, starting with a preload of 15% up to 85% of its RTS; with 15–20% increments, each new load returns to the preload level. The step duration is 3600 s.

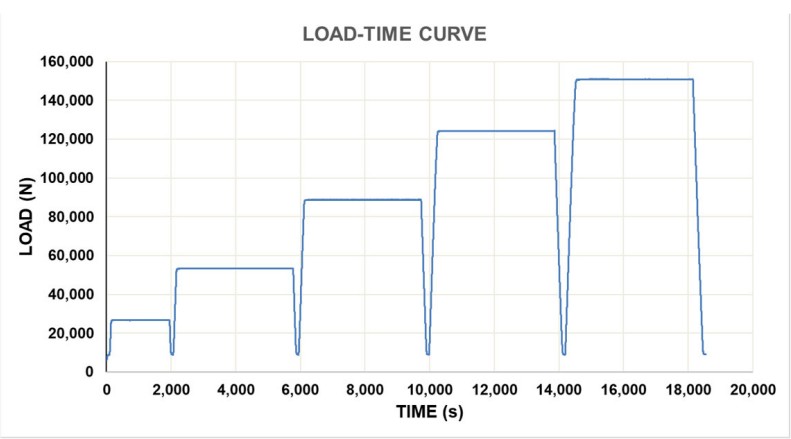

**Figure 17.** Stress-strain test for overhead conductors.

## 4. Discussion

The present work shows the feasibility of building a horizontal tensile machine to test specimens up to 12 m in length. The design can permanently contain the device for the suspension clamps and support loads up to 300 kN.

The analytical and numerical methods for the conductor test show that the stress results have a safety factor superior to 1.5, varying by 12%, and adding the deformation values supports the above statement.

Also, it is possible to provide evidence that the plate-fork fastening system is feasible to attach to the load cell and the traveling carriage to test the other products that usually are tested with MTV. Due to the specimen's length, another essential characteristic implemented is the alignment of longitudinal axes between the mobile carriage and the load cell, ensuring correct readings. Furthermore, a no-less-critical characteristic is the arrangement of the wheels of the mobile carriage, avoiding collapse against the rails during the performance test and ensuring no damage and long-term operation.

## 5. Patents

PATENT: Antonio Ramírez Martínez, Título de Modelo de Utilidad No. 4776, "Aditamento para Prueba de Grapas de Suspensión y Herrajes Integrado a Máquina de Tensión Horizontal". Clasificación; G01N3/08; G01N3/02, Instituto Mexicano de la Propiedad Industrial.

**Author Contributions:** Conceptualization, A.R.-M., L.B.-R., N.A.R.-O. and J.A.S.-C.; Data curation, A.R.-M.; Formal analysis, A.R.-M.; Funding acquisition, A.R.-M.; Investigation, A.R.-M., L.B.-R., N.A.R.-O. and J.A.S.-C.; Methodology, A.R.-M., L.B.-R., N.A.R.-O. and J.A.S.-C.; Project administration, A.R.-M.; Resources, A.R.-M. and N.A.R.-O.; Software, L.B.-R., N.A.R.-O. and J.A.S.-C.; Supervision, A.R.-M. and N.A.R.-O.; Validation, A.R.-M., L.B.-R., N.A.R.-O. and J.A.S.-C.; Visualization, A.R.-M.—original draft, A.R.-M.; Writing—review and editing, A.R.-M., L.B.-R., N.A.R.-O. and J.A.S.-C. All authors have read and agreed to the published version of the manuscript.

**Funding:** This research received no external funding.

**Conflicts of Interest:** The authors declare no conflict of interest.

## Nomenclature

| | |
|---|---|
| $A_b$ = Holes cross sectional area | $r$ = Radius of gyration |
| $\alpha$ = Suspension clamp dangle angle | K = Effective length factor |
| LRFD = Load and resistance factor design | b= Flange dimension |
| $A_n$ = Net cross-sectional area of member | $E$ = Modulus of elasticity |
| $A_g$ = Gross cross-sectional area of member | h = Web dimension |
| $\lambda$ = Width-to- Thickness Ratio | $F_e$ = Elastic buckling stress |

$P_n$ = Nominal compressive strength

$\lambda_r$ = Limiting Width-to-Thickness Ratio

$g$ = Transverse center-to-center spacing (gage) between fastener gage lines

$L_c = KL$ = Effective length of member

L = Laterally unbraced length of the member

$\varnothing_c$ = Load and Resistance Factor Design (LRFD)

$t_f$ = Flange thickness

$t_w$ = Web thickness

$F_y$ = Yield strength

$F_{cr}$ = Critical stress

$s$ = Longitudinal center-to-center spacing (pitch) of any two consecutive holes

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
