# Peer review of "Horizontal Tensile Machine for Mechanical Tests Applicable to Suspension Clamps, Transmission Line Accessories, and Overhead Conductors"

_machines, doi:10.3390/machines11050554_

Round 1

Reviewer 1 Report

Review 1

The article concerns an actual problem concerning the construction of a horizontal tensile machine. The paper presents a comprehensive presentation of the machine. Authors provided in the paper interesting solutions which can be used in many practical applications. Below, there have been expressed comments on which authors should improve the paper.

General remarks:

1.       Many figures, such as Figure 1, Figure 4, Figure 5, Figure 6, and Figure 7 should be enlarged. It would be good to present mentioned figures in better/higher quality.

2.       Figures 9-12 and 17-18 should be enlarged.

3.       The references should follow the Journal requirements

4.       It would be good to comment on the possibilities of applying the repeatable forces 

Author Response

By this means I send the corrections of the observations of the paper´s review   "Horizontal Tensile Machine for Mechanical Tests applicable to Suspension Clamps, Transmission Line Accessories, and Overhead Conductors".

Reviewer 2 Report

The current paper reports the development of a horizontal tensile machine for mechanical tests. The main drawback of the current paper is that, it is cluttered in the present form. There are too many short paragraphs without any cohesion among them. It is better to re-structured the whole paper accordingly. It is important to note that, this is a manuscript for scientific publication and not just a technical documentation. Based on my assessment, I suggest major revision of this paper.

The other comments are as follows:

1.     The abstract must be re-written. Try to keep it short and focused together with the inclusion of both quantitative and qualitative values. The first sentence of the current abstract may be omitted. In the current form it looks like a part of introduction section!

2.     In abstract, there is no need for sample/experimental details. It will contain the essence of the current work.

3.     The introduction section also needs to be restructured. Currently, it is cluttered and too many short paragraphs without any cohesion among them. It is suggested to keep the similar content in the same paragraphs and maintain link among the paragraphs.

4.     Is Fig. 1 original? If not, it require a ref. and so on for the rest of the manuscript.

5.     Again, for the rest of the manuscript, there are too much short paragraphs without proper linking among them.

6.     The think Fig. 3 will be better to mention as table.

7.     There is no mention of Table 4 in the text.

8.     The tables need to be restructured. For example, why Table 6 is like that? It can be easily organised as three-row table. Same remarks for Table  and others.

9.     I am missing the correlation of results obtained from numerical analysis and experimental results.

10.  There is hardly any scientific discussion in the paper. It just contain some results without given proper scientific explanation against the results.

11.  There is also no conclusion section!

Author Response

I send a response to the observations derived from the review of the document.

Reviewer 3 Report

1)      Line 40, 61, 131 - There are several places in the article where there is an extra space between adjacent words

2)      Line: 11 Unnecessary left-hand parenthesis in the designation of the ACSR power cable type

3)      Figure 1  - Poor quality drawings, they look like scans from the documentation, I suggest posting larger drawings with higher resolution

4)      In general, the drawings in the publication are of poor quality. Could the authors do something about it? They look scanned, not drawn specifically for this article. Especially visible in fig. 1, 4, 5, 6, 7

5)      Table 5 - Drawings and dimension designations are too small. I suggest you improve the drawings.

6)      Line: 257 - What is the meaning of "de" in a sentence?

7)      Line 280: I'm not quite sure "drastically" is the right word, may use "significantly" ?

8)      Figure 10 - Change the uppercase letter of the word "deformation" to lowercase

9)      Figure 16-17-18  The graphs are not the same size, the axis descriptions have different font sizes. I suggest unifying them

10)  In "References": there is no uniform style of spelling, for example, only the first letter of the name is present or the full name is present, all authors' surnames are capitalized, in other places only the first letter of the surname is capitalized. For example:

Line 366:

Antonio Ramirez-Martinez, Leonardo Barriga-Rodriguez, Luis Govinda Garcia-Valdovinos, Noe Amir Rodríguez-Olivares 366 and Alvaro Sanchez-Rodriguez, CREEP TEST FOR OVERHEAD CONDUCTORS USING DEADWEIGHT ACTUATED 367 LEVER, DYNA Ingeniería e Industria, DYNA March- April Vol.97 nº2 DOI: https://doi.org/10.6036/10245

Line 378:

A.V. Babushkin, D.S. Lobanov, A.V. Kozlova, I.D. Morev, Research of the effectiveness of mechanical testing methods with analysis of features of destructions and temperature effects, Russian Fracture Mechanics School,2013 DOI: 10.3221/IGF-ESIS.24.09.

Line: 383

Kazuya MITSUI, Atsushi SATO, FLEXURAL ELASTIC BUCKLING STRESS OF BATTEN TYPE LIGHT GAUGE BUILT-UP 383 MEMBER, CIVIL AND ENVIRONMENTAL ENGINEERING REPORTS, September 2017 , DOI: 10.1515/ceer- 384 2017-0027

There is a similar variation in many places. The uniform spelling system is not maintained, the entry should be corrected and standardized.

11) The scientific publication should contain conclusions resulting from the analysis of the results of the conducted research. The article does not present separately the conclusions of the presented research. The article with the conclusions should be supplemented.

Author Response

I hereby send a reply to the observations made in the review of the article.

Round 2

Reviewer 2 Report

can be accepted in current form